# Glucose Uptake by Skeletal Muscle within the Contexts of Type 2 Diabetes and Exercise: An Integrated Approach

**DOI:** 10.3390/nu14030647

**Published:** 2022-02-03

**Authors:** Nicholas A. Hulett, Rebecca L. Scalzo, Jane E. B. Reusch

**Affiliations:** 1Department of Medicine, Anschutz Medical Campus, University of Colorado, Aurora, CO 80045, USA; nicholas.hulett@cuanschutz.edu (N.A.H.); rebecca.scalzo@cuanschutz.edu (R.L.S.); 2Rocky Mountain Regional Veterans Affairs Medical Center, Aurora, CO 80045, USA; 3Center for Women’s Health Research, Anschutz Medical Campus, University of Colorado, Aurora, CO 80045, USA

**Keywords:** type 2 diabetes, insulin resistance, glucose transport, skeletal muscle, exercise

## Abstract

Type 2 diabetes continues to negatively impact the health of millions. The inability to respond to insulin to clear blood glucose (insulin resistance) is a key pathogenic driver of the disease. Skeletal muscle is the primary tissue for maintaining glucose homeostasis through glucose uptake via insulin-dependent and -independent mechanisms. Skeletal muscle is also responsive to exercise-meditated glucose transport, and as such, exercise is a cornerstone for glucose management in people with type 2 diabetes. Skeletal muscle glucose uptake requires a concert of events. First, the glucose-rich blood must be transported to the skeletal muscle. Next, the glucose must traverse the endothelium, extracellular matrix, and skeletal muscle membrane. Lastly, intracellular metabolic processes must be activated to maintain the diffusion gradient to facilitate glucose transport into the cell. This review aims to examine the physiology at each of these steps in healthy individuals, analyze the dysregulation affecting these pathways associated with type 2 diabetes, and describe the mechanisms by which exercise acts to increase glucose uptake.

## 1. Introduction

Type 2 diabetes (T2D) prevalence continues to steadily rise in the United States, reaching over 10% of the adult population [1]. The rising prevalence is associated with significant morbidity and premature mortality [2]. Diabetes contributes to systemic complications including cardiac and vascular events, kidney failure, neuropathies, vision loss, and cancer [3]. These problems carry significant finical burden, one in four healthcare dollars in the United States alone in 2017, which is expected to increase to $2.1 trillion globally by 2045, outpacing predicted GDP by 22.2% [4].

Insulin-mediated skeletal muscle glucose uptake requires a concert of physiological events. Upon ingestion of nutrients, the pancreas senses glucose elevation, leading to glucose-dependent insulin secretion and suppression of glucagon secretion. Insulin must circulate to target tissues to bind and take effect. One aspect of insulin action often underappreciated is insulin-dependent stimulation of endothelial nitric oxide synthase (eNOS). eNOS augments blood flow to insulin-sensitive target organs, including the cardiac and skeletal muscle. As such, insulin actively regulates delivery of itself, nutrients, and oxygen to the skeletal muscle. In the skeletal muscle, glucose and insulin must leave the circulation and traverse the endothelium and extracellular matrix to cross through the cell membrane to stimulate or enter the myocyte. Glucose can enter the skeletal muscle via both insulin-dependent and insulin-independent glucose transporters, including facilitated glucose transporter members (GLUT 1, 3, 4, 5, 8, 10, 11, and 12) [5]. Once glucose enters the muscle, it is trapped via phosphorylation to glucose-6-phosphate. The fate of this intracellular glucose is to either be metabolized or stored as glycogen. Delivery, transport, and metabolism are all modulated by an exercise bout. In this review, we aim to first lay out how glucose transport, metabolism, and storage occur in healthy skeletal muscle; describe the changes in these processes with T2D with a special emphasis on perfusion; and finally, to explore how exercise acts within the context of T2D at each of these steps to improve glucose uptake. Overall, we are interested in how systemic insulin resistance intersects with skeletal muscle glucose disposal and metabolism in the context of an exercise stressor.

## 2. Role of Skeletal Muscle in Blood Glucose Regulation

Skeletal muscle plays a principal role in post-prandial glucose regulation. After ingestion, ~80% of glucose is taken up by skeletal muscle via insulin-dependent glucose uptake [6,7]. Insulin-dependent and -independent skeletal muscle glucose disposal requires (1) glucose delivery to the muscle from circulation, (2) glucose traversing the extracellular matrix to the cell membrane, (3) uptake via facilitative glucose transporters either constitutively on the cell membrane or translocated in response to insulin or exercise, and (4) glucose gradient to facilitate glucose transport modulated by intracellular glucose metabolism, as shown in Figure 1.

The initial step for skeletal muscle glucose clearance is delivery. As such, skeletal muscle blood flow and perfusion play a key role in glucose disposal, which is often overlooked. It is interesting to speculate that tissue delivery is why insulin has evolved to regulate eNOS-dependent vasodilation [8]. The endothelium expresses the insulin and insulin like growth factor 1 (IGF 1) receptors [9]. When these receptors are activated, eNOS is activated via the Phosphokinase B (Akt/PKB) pathway, leading to vasodilation [10]. Experimental disruption of insulin signaling to eNOS, either globally or specifically in the skeletal muscle, lowers whole-body glucose disposal [11]. This mechanism of insulin action appears to be specifically vital in skeletal muscle, as blocking insulin-induced vasodilation in adipose tissue does not impair glucose uptake [12]. Insulin-mediated augmentation of perfusion of the skeletal muscle provides access to insulin, glucose, and oxygen, all of which are essential for glucose clearance and metabolism.

When skeletal muscle is perfused, glucose can be cleared from the circulation through the interstitial space and into the skeletal muscle. The extracellular matrix of healthy skeletal muscle is free of inflammation and fibrosis, allowing for rapid diffusion and transport of insulin and glucose across the endothelium and basement membrane to the skeletal muscle [13]. In the fasted state, the skeletal muscle is exposed to low insulin levels. In this state, glucose transport in skeletal muscle is facilitated via constitutive transporters on the skeletal muscle membrane [14]. After feeding, circulating insulin levels increase, insulin binds to its skeletal muscle receptor and signals GLUT translocation to the membrane. Briefly, insulin binds the insulin receptor causing phosphorylation of the insulin receptor substrate, which then activates the Akt/PKB pathway. Activation of the Akt/PKB pathway triggers the translocation of GLUT4 from the cytosol into the membrane, allowing glucose to move down its concentration gradient into the cell [15]. Of note, there is a complex post-receptor interplay of multiple signaling events in the glucose transport signaling pathway. This process, discovered more than 30 years ago, is now understood in exquisite detail, and has recently been reviewed [16,17,18,19]. GLUT4 is the main GLUT isoform that translocates to the cell membrane with insulin stimulation; however, GLUT 12 has also been shown to embed, as well [20].

Insulin is not unique in its ability to enhance skeletal muscle glucose transport. Contraction of skeletal muscle also signals GLUT4 translocation. Multiple upstream mechanisms including Rac1/actin and Ca^2+^/calmodulin-dependent protein kinase (CAMK) signaling are activated via muscle contraction [21]. These two signaling networks share downstream key proteins such as AMP-activated protein kinase (AMPK) and PI3K while also relying on novel targets such as TBC1D1/4 [22,23]. While GLUT4 is the most highly expressed glucose transport protein in skeletal muscle, there are other GLUT isoforms with physiological significance, notably GLUT1, which is constitutively present on the surface of skeletal muscle [24]. When GLUT1 is experimentally increased, basal glucose uptake also increases [25]. Glucose has also been shown to stimulate its own uptake through a process known as “glucose effectiveness” [26]. Each GLUT isoform is a facilitated transporter requiring a diffusion gradient to drive glucose into the cell.

Once in the cell, glucose must be utilized both to meet metabolic demands and to maintain the concentration gradient for facilitated transport. Glucose is initially phosphorylated via hexokinase, which traps the glucose intracellularly in the skeletal muscle, where it will succumb to one of four main fates: storage as glycogen, substrate in glycolysis, substrate for protein synthesis via the hexosamine pathway, or substrate in the pentose phosphate pathway [27]. Selection of glucose fate is dependent on the metabolic demands of the cell. In times of low energetic demand, glucose can be stored as glycogen, a highly branched polysaccharide. Glycogen storage is limited in skeletal muscle and therefore regulated by negative feedback on glycogen synthase by glycogen. Phosphorylated glucose can also enter glycolysis, a key ATP-producing pathway which also supplies fuel for the Krebs cycle and oxidative phosphorylation. This pathway is also negatively regulated by its end products, which accumulate during times of low energetic demand. The hexosamine pathway uses an intermediate product of glycolysis, fructose-6-phosphate, to produce nucleotides needed for protein synthesis [28]. Lastly, the pentose phosphate pathway is responsible for nicotinamide adenine dinucleotide phosphate, ribose 5-phosphate, and erythrose-4-phosophate production, which are critical factors for anabolism [29]. Flux through these pathways is driven by the needs of the cell in a dynamic manner. Each of these pathways can become saturated with excess glucose influx.

When skeletal muscle is adequately supplied with nutrient rich blood, has sufficient membrane permeability to glucose, and can maintain a diffusion gradient by storage and metabolism, large amounts of glucose can be cleared, maintaining systemic carbohydrate homeostasis. Skeletal muscle is a primary target of systemic glucose disposal; thus, interference in skeletal muscle glucose transport is sufficient to induce whole-body insulin resistance [30,31]. Loss of skeletal muscle glucose uptake is associated with abnormal carbohydrate metabolism, T2D, and associated with the development of atherogenic dyslipidemia [32]. While debated, there is evidence in humans that skeletal muscle acquires insulin resistance before adipocytes and hepatocytes [32].

## 3. Changes in Skeletal Muscle Physiology in Type 2 Diabetes

Skeletal muscle in people with T2D demonstrates diverse pathological changes impacting the delivery, uptake, and metabolism of glucose. The ability of skeletal muscle to respond to insulin is a primary factor in the pathophysiology associated with T2D. This phenotype likely has several converging mechanisms of action whose relative contributions remain contentious. Briefly, in individuals with a genetic predisposition for T2D, chronic nutrient overload, physical inactivity, and subsequent obesity lead to excess and ectopically stored lipid. The species of lipid, the cellular location, and turnover rates all influence insulin sensitivity [33]. For example, the increased diacylglycerol concentration within skeletal muscle associated with overnutrition and sedentary behavior leads to inflammation, oxidative damage, and fibrosis, reducing skeletal muscle’s ability to respond to demand [34]. Alternative activation of serine/threonine kinases and subsequent decreases in insulin receptor and IRS-1 tyrosine phosphorylation have also been implicated in disruptions of insulin signaling [35]. The skeletal muscle adapts (or maladapts) to chronic nutrient excess with changes in vascular structure and function—including endothelial dysfunction and accumulation of matrix proteins—and alterations in insulin signaling, and then reshapes the metabolic character, all of which contribute to reduced skeletal muscle glucose uptake, as shown in Figure 2 [36].

An objective of this review is to highlight a body of data on the contribution of abnormal microvascular structure and function to insulin resistance and hyperglycemia [37]. Microvascular complications are a hallmark of T2D-specific complications, including cardiomyopathy, nephropathy, retinopathy, and neuropathy. Changes in microvascular structure and function are present in the skeletal muscle, heart, brain, and islet in diabetes [38]. These microvascular changes suggest a role of the vasculature of insulin resistant skeletal muscle. The endothelium lining of the vasculature is responsible for sensing and responding to oxygen and nutrients within the blood [39]. As mentioned above, insulin and glucose are both signals to the endothelium to vasodilate to increase skeletal muscle perfusion and associated delivery and disposal of glucose [40]. However, these signals are disrupted with insulin resistance. T2D causes systemic endothelial cell damage through reactive oxygen and nitrogen species with reduced antioxidant potentials, reducing the cell’s ability to respond to stimuli [41]. For example, nitric oxide production is lower in response to insulin in insulin resistance, resulting in improper perfusion of skeletal muscle and subsequent impairments of insulin, glucose, and oxygen delivery [40,42,43]. Insulin resistance can be partially prevented when vasodilators are given, demonstrating the importance of blood flow for insulin action [44].

The endothelium plays a role in skeletal muscle insulin delivery in the context of T2D. There is debate in the field regarding the primary mechanism by which insulin moves from circulation to skeletal muscle. The two main positions are that insulin travels through the endothelial cell via transport proteins or between cells via fluid-phase transport. In support of insulin traveling through the cell is the observation that insulin accumulates in the endothelial cells at five times the concentration compared with circulation, requiring PI-3 kinase and mitogen-activated protein kinase kinase (MEK-kinase) signaling for proper transport to skeletal muscle [40,45]. Additional reports demonstrate that most insulin leaves circulation via fluid-phase transport and is independent of binding in endothelial cells [42,46]. Hyperglycemia associated with insulin resistance negatively affects insulin delivery to skeletal muscle and directly injures the endothelium [47]. Hyperglycemia causes the formation of advanced glycation end products that lead to proinflammatory signaling in the vessel walls and further reduce endothelial function [48].

Glucose and insulin access to the skeletal muscle are also limited by T2D-mediated changes to the perivascular and skeletal muscle extracellular matrix. The extracellular matrix plays a key role in capillary angiogenesis and regression [49] and insulin signaling [50]. In a pro-inflammatory state, as in T2D, skeletal muscle collagen production and turnover is dysregulated. Extracellular matrix remodeling is necessary for new capillary growth. Dysregulation of matrix remodeling can decrease capillary density and also impede the effective transfer of material from the bloodstream to the skeletal muscle. In T2D, there is an expansion of the extracellular matrix and decreased capillary density [13]. Hyaluronan, a chief element of a structural component of the extracellular matrix, is increased in those with T2D. Insulin action improves when hyaluronan expression is experimentally decreased in high fat fed mice [51]. Collagen deposition is also commonly associated with T2D [50]. Greater collagen is associated with lower muscle matrix metallopeptidase 9 activity in skeletal muscle [52]. Integrins, a class of matrix proteins, actively induce insulin resistance. For example, deletion of integrin-linked kinase from the skeletal muscle of mice had no effect on insulin action in lean mice. In contrast, when the same gene was deleted in obese mice, capillaries and insulin action were greater [53]. Manipulation of downstream molecules such as focal adhesion kinase demonstrate similar results [54]. These data demonstrate important barriers to skeletal muscle insulin delivery and glucose transport upstream of direct insulin action in T2D skeletal muscle.

Changes in skeletal muscle insulin signaling and cellular context occur in T2D. For example, GLUT proteins must be present in the membrane for facilitated glucose transport. In the context of T2D, GLUT4 content and translocation to the membrane is impaired due to insulin resistance. Increasingly, the problem of overnutrition and nutrient load on skeletal muscle has come to the forefront of understanding insulin resistance. Chronic overnutrition results in increased glycerol and free fatty acids in circulation. Normally, these nutrients are widely cleared by skeletal muscle and adipose tissue. Free fatty acids can diffuse into the cell or are taken up via a transporter such as CD36 and stored as intramyocellular lipids. However, excess in intramuscular triglyceride stores have been linked by numerous studies to insulin resistance [55,56,57]. When there is buildup of lipids, toxic metabolites such as diacetyl glyceride, ceramide, and acylcarnitine can accumulate. These metabolites act as signaling molecules that alter proteins and signaling cascades involved in insulin signaling, oxidative phosphorylation, and peroxisomal metabolism in skeletal muscle [58]. In addition, many of these lipid species increase the action of protein kinase Cθ, which phosphorylates serine residues of the insulin receptor and IRS, downregulating their action [59]. Similarly, toxic lipids drive inflammatory processes associated with insulin resistance [60,61]. In the context of excess nutrients, a high protonmotive force and NADH/NAD+ ratio occurs in the mitochondria as fuels are oxidized [62]. These conditions, without a constructive means to dissipate the protonmotive force (such as ATP production driven by demand), create a high H_2_O_2_ emitting potential [63,64]. The subsequent free radicals are proposed to further exacerbate insulin resistance [36]. The contribution of oxidative stress to insulin resistance is supported by data showing that treating high fat fed mice with a mitochondrial-specific antioxidant preserves insulin sensitivity [65].

Pathologic intracellular changes in skeletal muscle cell metabolism in T2D create a viscous cycle exacerbating insulin resistance in the context of overnutrition. As mentioned above, glucose enters the cell via facilitated transport down a diffusion gradient. To phosphorylate glucose, hexokinase must be present and active, and its product, glucose-6-phosphate, must be further metabolized to prevent allosteric down-regulation. When these processes are saturated, glucose-6-phosphate concentration increases and inhibits hexokinase activity, decreasing glucose transport. The concentration and compartmentalization of hexokinase are changed in T2D. Insulin signaling increases hexokinase [66]. Due to the relative decrease in insulin signaling with T2D, hexokinase expression levels are approximately 80% lower compared with controls [67]. Hexokinase overexpression in rats leads to improved insulin sensitivity [68]. In T2D, each of the pathways for intracellular glucose metabolism can become saturated. Glycogen storage reaches maximum capacity quickly during overnutrition. Glycolysis is allosterically regulated in the presence of the high levels of ATP. Allosteric down-regulation of glycolysis in T2D can increase intermediates such as methylglyoxal [69]. These highly reactive molecules bind to proteins and alter function [70]. This creates dicarbonyl stress, which those with T2D are less able to respond to due to a decrease in glyoxalase-1 [71]. Amino acid synthesis is also allosterically regulated by its end products. Lastly, de novo lipogenesis is limited as the cell becomes unhealthy with ectopic lipid accumulation. New research in T cells has provided a novel method where metabolic substrate flux can control transcription [72]. Given the shift in substrate metabolism within both T2D and exercise, there may be a role for metabolic enzymes as RNA-binding proteins.

## 4. Skeletal Muscle’s Response to Exercise within the Context of Type 2 Diabetes

Exercise is a cornerstone treatment for T2D due to both the acute and chronic improvements in skeletal muscle glucose uptake. Acute exercise is an energetic stress that requires upregulation of glucose delivery, uptake, and metabolism in skeletal muscle including glycolysis, the Krebs cycle, and oxidative phosphorylation. As discussed above, T2D creates profound changes in skeletal muscle, some of which modify the response to an exercise challenge as shown in Figure 3.

Exercise requires increased blood flow to skeletal muscle to meet oxygen and nutrient requirements. In the absence of T2D, maximal blood flow and peak oxygen uptake (VO_2peak_) are tightly correlated, and cardiorespiratory fitness is largely limited by central supply constraints [73]. In adults with T2D, modest improvements in cardiac function did not alter VO_2peak_, consistent with a contribution from skeletal muscle factors [74,75]. Skeletal muscle blood flow is heterogeneous during exercise in T2D, which results in lower skeletal muscle oxygen extraction [76]. Our group demonstrated that this heterogeneous blood flow aligns with poor oxygen extraction that is not due to decreased total limb blood flow or differences in hematocrit [77]. To determine the role of skeletal muscle oxygenation in metabolic function, we measured skeletal muscle oxidative phosphorylation with and without supplemental oxygen in adults with T2D and in healthy controls. Oxidative phosphorylation was lower in people with T2D compared with controls and improved with supplemental oxygen [78]. This response was absent in the control participants. Considering that mitochondrial function assessed ex vivo was not different between study populations, the data support a model wherein part of the muscle impairment in T2D is secondary to local perfusion and blood flow distribution limitations and lower oxygen extraction. These mechanisms are in part responsible for the lower VO_2Peak_ in people with T2D beyond modest impairments in the skeletal muscle mitochondria.

In addition to the benefits of a single exercise bout, exercise training also effects several factors influencing the delivery of glucose to skeletal muscle [79]. Exercise training is known to increase capillary density in those without T2D; however, the evidence for this phenomenon in those with T2D is mixed [80,81]. The impact of training on skeletal muscle oxygen delivery appears beneficial [82]. Exercise training in people with T2D does appear to remodel the extracellular matrix [83]. When accompanied with weight loss, exercise training improved insulin sensitivity more than weight loss alone. Collagen I and II deposition in the extracellular matrix of skeletal muscle was also lower with exercise. This change was associated with decreases in transforming growth factor β1 and Suppressor of Mothers Against Decapentaplegic 2/3 (SMAD 2/3) transcription. This is likely due to an increase in follistatin, an antagonist to transforming growth factor β1 [84]. In addition, a more fibrotic extracellular matrix is associated with lower cardiorespiratory fitness [85].

Exercise also impacts the uptake of insulin and glucose. Acute exercise increases energetic demands, particularly in skeletal muscle. These demands can be met through increased glycolysis and oxidative phosphorylation of ADP, which require greater insulin-independent glucose uptake [22,86]. In fact, those with T2D are capable of glucose uptake at levels near or identical to those without the disease in response to exercise [87]. Activation of multiple pathways can lead to insulin-independent GLUT4 translocation, as mentioned above. The energetic demand of exercise also stimulates the transport of glucose across the skeletal muscle membrane. A principal signaling mechanism in this process is AMP-activated protein kinase (AMPK), which can be viewed as an energy sensor within the cell. During exercise, large amounts of ATP are hydrolyzed, generating AMP and activating AMPK. Active AMPK is responsible for phosphorylating a wide variety of proteins within the cell, including AS160, which, when phosphorylated, signals for GLUT4 translocation [88]. Changes to intramuscular lipids and free fatty acids have been reported to occur in those with T2D [33]. There are also reported sex differences in intramuscular lipid utilization [89]. However, the effect of T2D in this context has not been investigated to this point. Differences between normal weight control subjects, individuals with obesity, and those with T2D highlight the need for specialized studies to answer questions regarding any change in skeletal muscle lipid content with exercise.

Chronic exercise training also affects factors related to glucose uptake. Insulin sensitivity is improved by exercise training [90]. A key protein for glucose uptake, GLUT4, has been shown to increase by 20–70% with aerobic exercise training [5]. A class of protein kinases known as calmodulin-dependent protein kinases have been shown in mice to positively regulate GLUT4 translocation in the context of exercise [21]. Fewer studies have examined the roles of other members of the GLUT family and their possible role. GLUT5 expression decreased by 72% following an exercise training program in sedentary individuals without T2D [91]. GLUT12 has also been measured with increased expression in a similar setting [92]. The contribution of these changes to skeletal muscle glucose uptake and if these findings translate to those with T2D is unknown. Effects of exercise training on intramuscular triglycerides of those with T2D is mixed, with studies showing increasing [93], decreasing [94], and stable levels [95]. However, levels of lipid and relative abundance of toxic versus neutral lipid species shifts with exercise training, which is suggested to provide an insulin-sensitizing effect independent of a repeated acute effect [33,96,97].

Exercise also has major effects on the metabolism of skeletal muscle influencing glucose uptake. Hexokinase expression and activity in skeletal muscle increase in response to exercise [98,99]. During exercise, glycogen stores are utilized as fuel for glycolysis. Chronically, exercise is known to increase glycogen storage capacities of skeletal muscle, providing a larger glucose sink. Studies examining the effect of aerobic exercise training on glycolytic capacity are mixed, with some showing increased levels while others show no change [5]. The reason(s) for this may be related to the type of skeletal muscle as well as the specifics of the exercise prescription (frequency, modality, and/or intensity). While the hexosamine pathway is associated with insulin resistance, very few studies have investigated the effect of an exercise bout on this pathway, and none included participants with T2D. Exercise training did not alter gene expression of key metabolic enzymes associated with the hexosamine pathway in postmenopausal women [100]. However, Sprague–Dawley rats did have greater enzyme concentrations after exercise training compared with sedentary rats [101]. More research is needed to determine the effect of exercise on this pathway and to probe any changes brought about by T2D.

There are likely other glucose-handling changes associated with exercise and T2D in skeletal muscle. Profound transcriptomic variations have been observed, including 819 differentially expressed genes in exercised versus sedentary rats [102]. Of particular interest is the protein glyoxalase-1, which reduces carbonyl stress and increased after 4 weeks of exercise training [103]. Differences in enzymes involved in the malate-aspartate shuttle, citric acid cycle, and glycolytic protein levels were measured in humans [103,104]. More research is needed to understand how these transcriptomic changes relate to changes in protein abundance and enzymatic activity. Even without a change in protein abundance, post-translational modifications associated with T2D adjust protein function. C-reactive proteins, which alter protein function, are altered in the T2D environment [105], as well as heat shock protein differences, which closely correlate with insulin resistance [106]. Mitochondrial biogenesis is associated with exercise in those with T2D [107]. However, this effect is not seen uniformly across the lifespan or between sexes. Estrogen may have differential effects on mitochondrial biogenesis dependent on T2D presence (unpublished data), and the impairment in peak oxygen consumption associated with T2D is greater in women compared with men [108].

Given the integrated structural and functional factors responsible for skeletal muscle glucose uptake, which are impacted by T2D and exercise, as outlined above, the adaptive responses to therapeutic interventions should be considered in the full in vivo context. Optimization of glucose metabolism in the skeletal muscle involves the augmentation of skeletal muscle glucose utilization that can be affected either by pharmacotherapy or exercise training. Improvements in skeletal muscle insulin action are impacted by nutrition, exercise, and pharmacotherapy. Augmentation of skeletal muscle capillary density and endothelial function can be achieved in response to exercise training, normalization of glycemia, and in response to pharmacotherapy therapy. Modulation of the proinflammatory state in T2D will impact all targets outlined in the review.

Skeletal muscle is increasingly thought of as an endocrine organ through the production and release of myokines and extracellular vesicles [3,109]. These myokines and extracellular vesicles have profound effects on other tissues, especially in the context of T2D [110]. The effects of IL6 and other myokines have been expertly reviewed and found to affect distant tissues and processes, including beta cell proliferation in the pancreas and glucose production in the liver [111]. Extracellular vesicles are small membrane-bound particles capable of targeted delivery of protein, RNA, and lipid cargo [112]. These vesicles provide a novel method of intercellular communication within the context of exercise. During a one-hour cycling bout, there was a ~2-fold increase in the number of extracellular vesicles containing over 300 differentially expressed proteins [113]. Many of these vesicles were tracked to the liver, a key organ in glucose homeostasis [113]. While determining the origin of these vessels is technically limited at this time, there is reason to suspect that skeletal muscle secretes a substantial number, as skeletal muscle comprises a large amount of mass in the body and contains the protein machinery required for bulk extracellular vesicle biogenesis. In fact, muscle releases ~80% more extracellular vesicles than white adipose tissue in vitro [114]. While extracellular vesicles have been implicated in T2D complications and suggested as a potential therapeutic, little is known about changes to their composition due to exercise in this population [110,115,116].

The type of exercise training performed is a critical determinant of the adaptive response. Resistance training has been included with aerobic training in physical activity guidelines for those with T2D. Yet, much less is known about the response of skeletal muscle to resistance training in the context of T2D. Resistance training is performed at a higher intensity and relies more on glycolytic means of energy production. A small study found that resistance training increased mitochondrial content [117]. Moreover, increased glucose extraction has been measured in resistance-trained versus untrained limbs of people with T2D [118]. This change in glucose extraction may be due to an improved microvascular response [119]. The specifics regarding a resistance training prescription warrants more research.

## 5. Conclusions and Future Directions

Skeletal muscle plays a major role in glucose homeostasis, and skeletal muscle insulin resistance is central to T2D. Delivery, uptake, and metabolism are necessary steps for skeletal muscle insulin action and glucose disposal. It is critical to address each one of these steps when considering the role of skeletal muscle in glucose regulation and its impairments in T2D. By neglecting less canonical factors such as blood flow and extracellular matrix changes, the model fails to reflect the physiological reality of in vivo insulin action. T2D entails changes in skeletal muscle insulin action, blood flow distribution, capillary density, and extracellular matrix, each of which can limit glucose disposal. Exercise and its insulin-sensitizing action must also be understood within in whole muscle and its modification in T2D. As an intervention, exercise acts to improve skeletal muscle’s insulin action and glucose clearance through signaling cascades, both within and outside the skeletal muscle cell itself. More research is needed to understand the mechanisms with differential response to exercise in the T2D environment and to identify differences compared to non-diabetic controls and between men and women. Of particular interest to our lab group is the intersection of the microvascular changes to skeletal muscle fuel partitioning and mitochondrial dynamics. The effect of sex differences within this context also remains to be explored.

## Figures and Tables

**Figure 1 nutrients-14-00647-f001:**
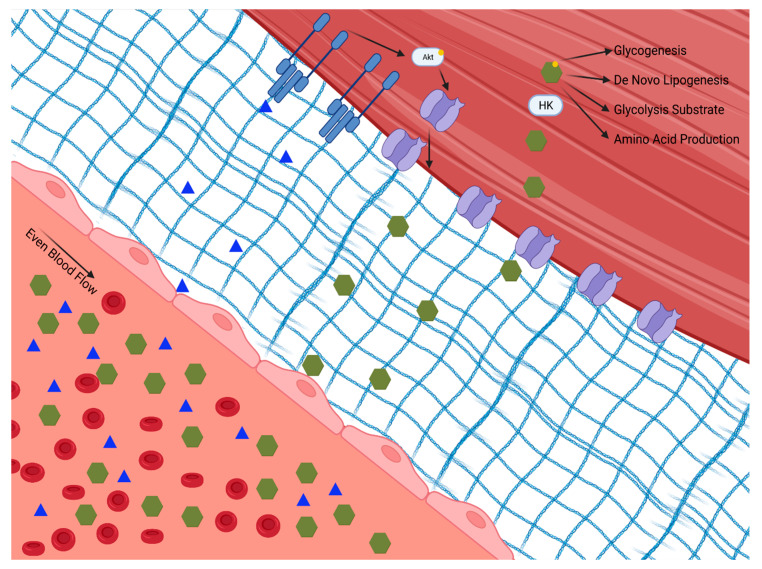
Insulin-dependent and -independent skeletal muscle glucose disposal requires: (1) glucose delivery to the muscle from circulation through the extracellular matrix to the cell membrane; (2) uptake via facilitative glucose transporters either constitutively on the cell membrane or translocated in response to insulin or exercise; and (3) a glucose diffusion gradient to drive glucose into the cell which is modulated by intracellular glucose metabolism. Hexokinase (HK). Phosphokinase B (Akt).

**Figure 2 nutrients-14-00647-f002:**
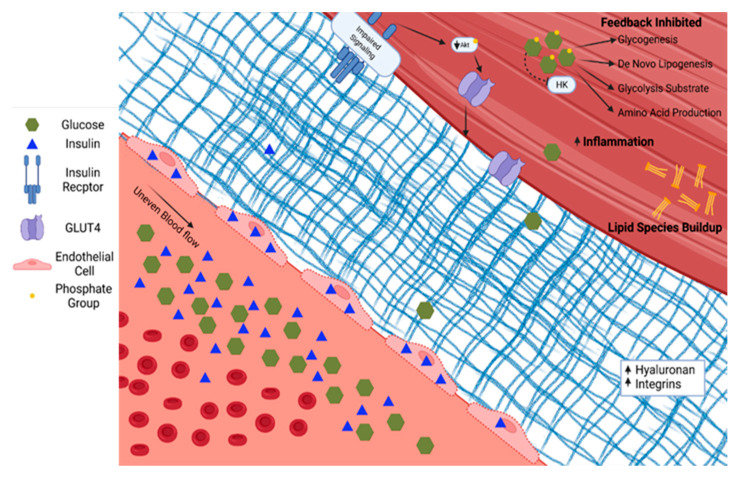
Type 2 diabetes is characterized by increased glucose and insulin in circulation. Insulin accumulates in endothelial cells. The extracellular matrix becomes fibrotic with increased hyaluronan and integrins. Serine/threonine phosphorylation on the insulin receptor and insulin response substrates leads to blunted insulin signaling through PI3K/Akt. The glucose diffusion gradient is limited by elevated intracellular glucose concentrations and allosteric down-regulation of intracellular glucose metabolism. Hexokinase (HK). Phosphokinase B (Akt).

**Figure 3 nutrients-14-00647-f003:**
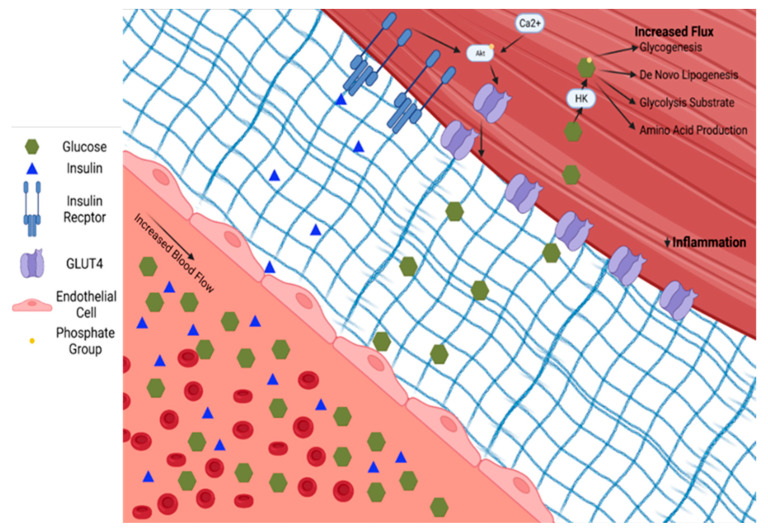
Exercise training restores proper blood flow to skeletal muscle and increases insulin. The extracellular matrix becomes less fibrotic, allowing the passage of glucose and insulin to skeletal muscle. Intramuscular glucose metabolism increased, thereby decreasing the allosteric downregulation of glucose disposal and augmentation of the glucose gradient for facilitated glucose transport. Decreased intracellular DAG and toxic lipid accumulation improves post-receptor insulin action. Hexokinase (HK). Phosphokinase B (Akt).

## Data Availability

Not applicable.

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
