# Peer review of "Glucose Uptake by Skeletal Muscle within the Contexts of Type 2 Diabetes and Exercise: An Integrated Approach"

_nutrients, 2022, doi:10.3390/nu14030647_

Round 1

Reviewer 1 Report

The author has done extensive literature review for the article. However, the article requires professional editing as well as major revision to strengthen the manuscript for publication. The headings are very generalized, so we suggest making it more specific and then divide the information accordingly. By doing this the article will look more organized as well as informative.

  1. Figure 2 and 3. The graphical representation are not very well defined. The author should compare the changes in muscle physiology or effect of exercise in control and type 2 diabetic conditions simultaneously. Different cells and receptors should be labelled or defined so that it is easy for the reader to understand.

  1. Line 140-141 - “A primary factor in the pathophysiology associated with T2D is the ability of skeletal muscle to respond to insulin .” It should be

  1. Author can improve the writing style of the review paper. Many repetitions of the information are confusing as a reader. There is no connection with the preceding statement and hence what information does the author wants to give seems vague. For instance, pls check lines 285-291. What kind of extracellular matrix remodeling is induced by exercise in T2D and how it connects with β1/SMAD 2/3 pathway?

  1. Line 274 - These values are disassociated only in individuals with T2D….” Please revise the statement as, these values should be associated with T2D.

  1. Line 281 - “exercise training effects on delivery to skeletal muscle….” By “delivery” what is author referring to ? Is it oxygen, glucose, insulin or other signaling molecules? Also, there is no reference to refer.  s

  1. Lines 303-305. What kind of lipid changes are being referred here? Where these changes occurs and what is their significance?

  1. Line 309-310 - Together these changes in response to exercise lead to an increase in insulin sensitivity of 18-30% that can remain for 48 hours”. Author should recheck the data with the referred paper. Cannot find these values in the reference. Also add some background information about the statement such as what study group and exercise training protocol was used to get these results.

  1. In the review paper author have discussed about the changes in muscle physiology during T2D. You can consider to add the recent paper where they propose dicarbonyl stress in skeletal muscle of T2D patients can also lead to the insulin resistance in muscle.

Mey, Jacob T., et al. "Dicarbonyl stress and glyoxalase enzyme system regulation in human skeletal muscle." American Journal of Physiology-Regulatory, Integrative and Comparative Physiology 314.2 (2018.

  1. Line 373 - “much less is known about skeletal muscle’s response to resistance training with T2D.” Author should refer to the following papers where they have observed “9 months of RT enhanced oxidation of both fatty acid and glycolytic derived substrates in skeletal muscle of individuals with T2D.”

Sparks, Lauren M., et al. "Nine months of combined training improves ex vivo skeletal muscle metabolism in individuals with type 2 diabetes." The Journal of Clinical Endocrinology & Metabolism 98.4 (2013)

  1. The author wants to focus on the role of microvascular changes to skeletal muscle in T2D conditions. Please refer to the Russell et.al 2017 paper and can include in the article as well.

Russell, Ryan D., et al. "Skeletal muscle microvascular-linked improvements in glycemic control from resistance training in individuals with type 2 diabetes." Diabetes Care 40.9 (2017).

  1. Line 272-273 - “The oxygen limitation in people with T2D was also observed when comparing in vivo and in vitro measures of mitochondrial function within the same individual.” The referred paper is performing the ex-vivo (biopsy sample) and in vivo (GCM) study from obese people with or without T2D. Kindly revise the statement.

Reviewer 2 Report

The authors present a review primarily discussing the response of skeletal muscle to exercise in Type 2 diabetes. The manuscript is very well-written and shows an adequate use of references. I have a few comments.

1- Could you please point out the novelty of this paper? The role of skeletal muscle in regulating blood glucose has been established in both normal and people with Type 2 diabetes. How is this paper adding to prior knowledge?    

2-Please add a paragraph discussing how understanding the mechanisms surrounding differential response to exercise will help plan future studies targeting exercise in type 2 diabetes. This is crucial to show that this might lead to implementing guidelines/recommendations in this population.

Minor : Typo line 29

            Lines 390-391: please rephrase

Round 2

Reviewer 1 Report

The manuscript is improved after revision. It is acceptable for publication.